# Trehalase Inhibitor Validamycin May Have Additional Mechanisms of Toxicology against *Rhizoctonia cerealis*

**DOI:** 10.3390/jof9080846

**Published:** 2023-08-14

**Authors:** Xiaoyue Yang, Yan Shu, Shulin Cao, Haiyan Sun, Xin Zhang, Aixiang Zhang, Yan Li, Dongfang Ma, Huaigu Chen, Wei Li

**Affiliations:** 1Institute of Plant Protection, Jiangsu Academy of Agricultural Sciences, Nanjing 210014, China; yangxy_1999@163.com (X.Y.); syan0306@foxmail.com (Y.S.); caoshulin@jaas.ac.cn (S.C.); sunhaiyan8205@126.com (H.S.); njuzhangxin@163.com (X.Z.); zhangaixiang2003@163.com (A.Z.); 2Key Laboratory of Sustainable Crop Production in the Middle Reaches of the Yangtze River, College of Agriculture, Yangtze University, Jingzhou 434025, China; vgly1987@sina.com (Y.L.); madf@yangtze.edu.cn (D.M.); 3Jiangsu Co-Innovation Centre for Modern Production Technology of Grain Crops, Yangzhou University, Yangzhou 225009, China

**Keywords:** *Rhizoctonia cerealis*, sharp eyespot of wheat, validamycin, trehalase, ribosome

## Abstract

Sharp eyespot is a crucial disease affecting cereal plants, such as bread wheat (*Triticum aestivum*) and barley (*Hordeum vulgare*), and is primarily caused by the pathogenic fungus *Rhizoctonia cerealis*. As disease severity has increased, it has become imperative to find an effective and reasonable control strategy. One such strategy is the use of the trehalose analog, validamycin, which has been shown to have a potent inhibitory effect on several trehalases found in both insects and fungi, and is widely used as a fungicide in agriculture. In this study, we demonstrated that 0.5 μg/mL validamycin on PDA plates had an inhibitory effect on *R. cerealis* strain R0301, but had no significant impact on *Fusarium graminearum* strain PH-1. Except for its inhibiting the trehalase activity of pathogenic fungi, little is known about its mechanism of action. Six trehalase genes were identified in the genome of *R. cerealis*, including one neutral trehalase and five acidic trehalase genes. Enzyme activity assays indicated that treatment with 5 μg/mL validamycin significantly reduces trehalase activity, providing evidence that validamycin treatment does indeed affect trehalase, even though the expression levels of most trehalase genes, except *Rc17406*, were not obviously affected. Transcriptome analysis revealed that treatment with validamycin downregulated genes involved in metabolic processes, ribosome biogenesis, and pathogenicity in the *R. cerealis*. KEGG pathway analysis further showed that validamycin affected genes related to the MAPK signaling pathway, with a significant decrease in ribosome synthesis and assembly. In conclusion, our results indicated that validamycin not only inhibits trehalose activity, but also affects the ribosome synthesis and MAPK pathways of *R. cerealis*, leading to the suppression of fungal growth and pesticidal effects. This study provides novel insights into the mechanism of action of validamycin.

## 1. Introduction

Sharp eyespot of wheat, caused by *Rhizoctonia cerealis* Van Der Hoeven, harms wheat, barley, and other cereal crops, and seriously affects crop yield and quality [1,2]. The disease is prevalent in warm and humid regions and is widely distributed in Europe, North America, Africa, Oceania, and Asia [1]. In China, sharp eyespots are widespread in nearly 20 provinces, especially in Jiangsu, Henan, Hebei, and other places [3]. Currently, the progress of breeding for resistance varieties against sharp eyespot of wheat is slow, and there are few varieties of wheat that are widely promoted for resistance to sharp eyespot [4]. Chemical control is the most commonly used method to prevent and control this disease [1]. At present, the main fungicides used to prevent sharp eyespots include validamycin, tebuconazole, triazolone, pyraclostrobin, thifluzamide, etc [5,6]. Among these, validamycin, as an antibiotic fungicide, has become an important and widely used fungicide for the prevention and control of wheat sharp eyespot and rice sheath blight, due to its high efficacy against diseases caused by *Rhizoctonia* fungi, as well as its lack of injury and pollution [7,8].

Validamycin has been intensively used to control sharp eyespots in China and Japan for two decades [9]. Previous studies have shown that validamycin, as the structural analog of trehalose, could competitively inhibit the activity of trehalase, thereby affecting the glucose metabolism of fungi [10,11,12,13,14,15]. Trehalase (EC.3.2.1.28) is an α-glucosidase hydrolase that can convert trehalose to glucose in fungi, insects, and mammals [16]. Currently, validamycin is mainly used for the prevention of sharp eyespot of wheat and sheath blight of rice caused by *Rhizoctonia* spp. (basidiomycetes), but its effect on controlling Fusarium head blight (FHB), caused by *Fusarium* spp. (ascomycetes), in wheat is not ideal [17]. However, recent research has shown that validamycin can be combined with DMI fungicides to reduce *Fusarium* mycotoxins production and prevent FHB [17,18,19,20]. However, the toxicology mechanisms of validamycin in fungi are unclear.

In previous studies, we found that validamycin significantly reduced the growth rate of *R. cerealis*, but had no significant inhibitory effect on *F. graminearum*. To our knowledge, no study has reported why validamycin has different inhibitory effects on basidiomycetes and ascomycetes. The aim of this study was to clarify the mechanism of validamycin in *R. cerealis* through enzyme activity determination and transcriptome data analysis and to explore whether there are other antifungal mechanisms of action in addition to inhibiting the activity of trehalase.

## 2. Materials and Methods

### 2.1. Strains, Culture Conditions, and Validamycin A Sensitivity Assay

*R. cerealis* strain R0301 and *F. graminearum* strain PH-1 were cultivated on potato dextrose agar (PDA) medium at 25 °C for 3 days. Validamycin A (60%) (Qianjiang Biochemical, Haining, China) was dissolved in sterile water. Growth tests were performed on PDA [20]. The culture medium for different carbon conditions was prepared by substituting glucose in the PDA medium with trehalose (PTA) to obtain media with the same carbon content (2% *w/v*) and different carbon sources. R0301 was treated with validamycin at 0, 0.5, 1, 2, 4, and 8 μg/mL concentrations, and PH-1 was treated at 0, 5, 20, 80, 160, and 320 μg/mL on different carbon source plates. Mycelial plugs, 5 mm in diameter, were placed on fresh plates of the above media and cultured for 3–8 days at 25 °C.

### 2.2. Extraction of Trehalase from the Mycelium of R0301

The total proteins from the mycelium were extracted by ammonium sulfate precipitation with modifications [21]. R0301 was cultured on PDA and then cut into mycelial pellets. These mycelial pellets were inoculated into potato dextrose broth (PDB) medium for 7 days at 25 °C on a reciprocating shaker. After incubation, the mycelia were collected by filtration and washed with CZAPEK’s salt solution. The mycelial mat was homogenized in a blender for 30 s with CZAPEK’s salt solution. The homogenized mycelia were filtered and washed with the same solution. The washed mycelia (120 g in wet weight) were suspended in 400 mL of 10 mM PBS (pH 6.0) and blended for 2 min. The suspension was sonicated for 20 min and then centrifuged at 3500 rpm for 20 min. Ammonium sulfate was added to the supernatant to achieve 40% saturation, and the precipitate was removed by centrifugation at 3500 rpm for 20 min. Furthermore, ammonium sulfate was added to the supernatant to 60% saturation, and the precipitate was harvested by centrifugation at 3500 rpm for 20 min and then suspended in a small volume of 1 mM PBS (pH 6.0). This solution was then dialyzed against 1 mM PBS (pH 6.0) at 4 °C overnight. The dialyzed solution was centrifuged at 3500 rpm for 5 min, the precipitate was obtained and dissolved in a small volume of ddH_2_O, and the extract of total proteins served as the source of trehalase.

### 2.3. Determination of Trehalase Activity in Mycelia

Trehalase activity was determined using the 3,5-dinitrosalicylic acid method with modifications [22]. The reaction mixture consisted of 125 μL of extract of total proteins, 50 μL of 0.4 M trehalose (Shanghai Yuanye Biotechnology Co., Ltd., Shanghai, China), 125 μL of 0.2 M PBS (pH 6.0), 200 μL of validamycin (final concentrations of 0.5, 1, 2, 5, 10, and 20 μg/mL), and sterile water as control (CK). The mixture was incubated at 37 °C for 15 min, and trehalase activity was then determined. The reaction was stopped by adding 500 μL 3,5-dinitrosalicylic acid and boiled water for 5 min, and then the optical density at 540 nm (OD_540_) was measured after cooling. Trehalase activity was determined by measuring the content of glucose released during incubation. One unit of enzyme (U) was defined as the amount that hydrolyzes 1 μmol of trehalose per minute.

### 2.4. Phylogenetic Analysis of Trehalase Genes in R. cerealis

Six homologs of trehalase genes were found from the genome of R0301 [23]. According to the protein sequences of target genes, homologous protein sequences of other species were obtained in the GenBank database, and some trehalase genes and homologous sequences were retrieved from published articles. The obtained protein sequences were compared using the MAFFT online tool (https://www.ebi.ac.uk/Tools/msa/mafft (accessed on 12 April 2023)), and then the phylogenetic tree was constructed by the maximum likelihood (Bootstrap Replications = 1000, Jones‒Taylor‒Thornton (JTT) model) method in Molecular Evolutionary Genetics Analysis (MEGA X 10.0.5) software [24]. Furthermore, the phylogenetic tree was modified via the online tool iTOL (https://itol.embl.de/itol.cgi (accessed on 12 April 2023)).

### 2.5. RNA Extraction and cDNA Synthesis

Strain R0301 was cultured at 25 °C on PDA, and after 3 days, it was reinoculated on PDA and PDA containing 0.5 μg/mL validamycin and incubated for 3 days, and mycelia were collected on the 4th day, ensuring 3 biological replicates per treatment. Total RNA was extracted from the vegetative hyphae of R0301 using RNAiso Reagent (TAKARA, Dalian, China). RNA was treated with Recombinant RNase Inhibitor (TAKARA, Dalian, China) and then reverse transcribed to cDNA with a HiScript II 1st Strand cDNA Synthesis Kit (Vazyme, Nanjing, China).

### 2.6. Construction of Transcriptome Libraries and Acquisition of Transcriptome Data

The raw data sequences were filtered to remove splice sequences and low-quality reads to obtain clean data, which were analyzed by FastQC (http://www.bioinformatics.babraham.ac.uk/projects/fastqc (accessed on 24 Nivember 2021)) for quality control. All six samples yielded more than 6 Gb of data. Hisat2 (version 2.1.0) was used to compare the high-quality sequencing data with the reference genome sequence of R0301 [23,25], and then transcript assembly was performed with StringTie (version 2.0) [26]. DESeq2 (version 1.26.0) software was used to analyze the differential gene expression between samples, with selection criteria of |log2FoldChange| ≥ 2 and FDR < 0.05 [27]. The differentially expressed genes were matched to the GO, KOG and KEGG databases, and differential expressed genes (DEGs) were analyzed for functional annotation and pathway enrichment.

## 3. Results

### 3.1. Validamycin Inhibited the Hyphal Growth and Trehalase Activity of R. cerealis

First, we determined that the optimum temperature was 50 °C and the optimum pH was 8.0 for the trehalase reaction in vitro using the 3,5-dinitrosalicylic acid method. Under these conditions, we determined the effect of different concentrations of validamycin on trehalase activity in the mycelium of *R. cerealis*. Compared with CK, the activity of trehalase in mycelium decreased after adding validamycin. When 5 µg/mL validamycin was added to PDB medium, the trehalase activity was the lowest. The value was only 0.64 U/mg, which significantly differed from that of CK (Figure 1a).

The colony morphologies of R0301 and PH-1 strains grown on solid medium with different carbon sources were compared. For *R. cerealis*, the inhibitory effect was obvious when the concentration of validamycin was low and more distinct when trehalose was added to the medium than when glucose was added (Figure 1b). For *F. graminearum*, the inhibitory effect was not obvious when the concentration of validamycin was high, and there was no significant difference between morphologies with the addition of glucose or trehalose in the medium (Figure 1c).

### 3.2. Trehalase Genes in R. cerealis and Their Expression after Validamycin Treatment

In the phylogenetic tree, the trehalases from insects and bacteria each cluster in one branch, while the remaining fungi cluster into three branches. Apart from some ascomycetes’ unique acidic trehalases, which cluster separately, the acidic trehalases of ascomycetes and basidiomycetes cluster together in one branch, while the neutral trehalases cluster in another branch. The five trehalases of *R. cerealis* are in the acidic trehalase branch, whereas one is in the neutral trehalase branch, leading to the inference that *Rc20849*, *Rc17406*, *Rc16440*, *Rc16514*, and *Rc16513* are acidic trehalase genes and *Rc10975* is the neutral trehalase gene (Figure 2a).

According to the analysis of differences in gene expression levels, after adding validamycin, *Rc16514*, *Rc16513* and *Rc20849* showed an increase, whereas *Rc10975*, *Rc17406* and *Rc16440* showed a decrease. Among these genes, the ones that were significantly increased and which decreased the most were *Rc16514* and *Rc17406*, respectively (Figure 2b). Although *Rc16514* expression was highly upregulated, its original expression level was low; meanwhile, *Rc17406* expression was highly downregulated, and its original expression level was high (Figure 2c). Therefore, the acid trehalase *Rc17406* may play a major role in trehalose degradation.

### 3.3. Integrated Analysis of the Transcriptome of R. cerealis

A total of 41.78 Gb of data were obtained, and the clean data of each sample reached more than 6.79 Gb; the percentage of Q30 bases was more than 93.33%. The data quality was determined, and the data could be used for subsequent functional analysis. In total, 6240 genes were detected by comprehensive sample expression analysis, including 3497 upregulated genes and 2743 downregulated genes (Figure 3a).

The transcriptome data were compared with the KOG database (Clusters of orthologous groups for eukaryotic complete genomes). Functional predictions were conducted, and the results showed that 24 functional categories were annotated. Among these, general function prediction (459) was the most common, followed by posttranslational modification, protein turnover, chaperones (206) and signal transduction mechanisms (199) (Figure 3b).

GO enrichment analysis was performed for upregulated and downregulated DEGs in transcriptome data treated with validamycin and data without special treatment (Figure 3c,d). Compared with the control group, the downregulated genes of the strains treated with validamycin were significantly enriched in various metabolic processes, including rRNA processing, ribosome biogenesis, ribosome and translation, and significantly enriched in pathogenicity (Figure 3d).

### 3.4. KEGG Enrichment Analysis of DEGs

KEGG pathway analysis of upregulated and downregulated DEGs showed that 1250 DEGs were concentrated in 24 pathways, and the differentially expressed pathways were mostly involved in metabolism and translation. Among the DEGs, 588 genes were upregulated, and 662 genes were downregulated (Figure 4a).

According to the enrichment results of the KEGG pathway, genes related to the MAPK signaling pathway were most affected, and the expression levels of 124 genes were changed among all 502 genes (24.7%) (Figure 4b). Among the remaining pathways, ribosome synthesis and assembly related pathways decreased significantly (Figure 4c).

In the ribosome biosynthesis pathway, 59 out of 149 genes (39.60%) had changed expression, and the vast majority were downregulated (Figure 4c). The expression of 89 of 206 genes (43.20%) related to ribosome composition changed, and only one gene was upregulated, while all the others were downregulated (Figure 4c). Therefore, it is speculated that treatment with validamycin not only inhibited the trehalase activity, but also inhibited the related functions of ribosomes, which led to altered protein translation and, thus, inhibited the growth and infestation of the fungi.

We used the FPKM expression values of genes in the above pathway to generate heatmaps, which showed that trehalase was significantly inhibited after validamycin treatment (Figure 5a), but many genes in the ribosome and ribosome biogenesis pathway were also significantly inhibited (Figure 5b,c).

## 4. Discussion

Validamycin, an aminoglycoside antibiotic, is produced by *Streptomyces hygroscopicus* 5008 [28] and is commonly used to control sharp eyespot of wheat caused by *R. cerealis* and sheath blight of rice caused by *R. solani* [1,7,8]. Previous studies have reported that validamycin could inhibit trehalase activity and alter energy metabolism, thereby inhibiting the growth of fungi and insects [12,13,15,29,30].

Recent studies have shown that validamycin inhibits deoxynivalenol (DON) biosynthesis by targeting the neutral trehalase from *F. graminearum* and reducing the interaction between neutral trehalase and pyruvate kinase, a key enzyme in glycolysis; furthermore, validamycin synergizes with demethylation inhibitor (DMI) fungicides against *F. graminearum* by decreasing sterol 14α-demethylase enzyme (CYP51s) expression [17,18,19,20]. In this study, we demonstrated that validamycin obviously inhibited the mycelium growth of *R. cerealis* on PDA plates at a concentration of 0. 5 µg/mL but had no obvious inhibitory effect on the mycelial growth of *F. graminearum* except at higher concentrations (Figure 1), which is consistent with a previous study [19]. This significant difference in antifungal activity prompted us to explore the mechanism of action of validamycin against *R. cerealis*. Although there are reports that validamycin reduces the progression of the disease by inhibiting trehalase activity, its target or mechanism of action is still unclear.

Then, we preliminarily studied the mechanism of validamycin to trehalase in *R. cerealis*. From the *R. cerealis* genome, six trehalase genes were obtained (Figure 2a). The results of transcriptome analysis showed that all 6 trehalase genes were affected after validamycin treatment, and the expression level of *Rc17406* was most downregulated, suggesting that this gene may play a major role in the metabolism of trehalose in *R. cerealis* (Figure 2b,c). However, the results of transcriptome analysis showed that only the expression of trehalase genes involved in trehalose synthesis and metabolism pathway was affected by validamycin treatment in starch and sucrose metabolism, while the expression of other genes remained unchanged (Figure 4). Unexpectedly, the expression of many genes associated with ribosome synthesis and assembly were suppressed by validamycin (Figure 5).

Ribosomes are universally conserved macromolecular machines that consist of a 40S small subunit and a 60S large subunit, which, together with translation factors, tRNAs and mRNA, catalyze the synthesis of proteins in every cell [31,32]. The assembly of eukaryotic ribosomes is a complicated and dynamic process that involves many preribosomal assembly intermediates. More than 200 trans-acting assembly factors (AFs) and many small nucleolar RNAs (snoRNAs) function in the modification and processing of ribosomal RNA (rRNA) and the assembly of ribosomal proteins (RPs) [33,34]. In particular, the ribosome plays a central role in RNA translation and is a prime target for antibiotics used in both clinical and agricultural settings [35,36]. Among the downregulated DEGs enriched in the transcriptome, there were 312 genes related to rRNA, 82 genes related to mRNA, 71 genes related to tRNA, and 59 genes related to snoRNA. Within these, we identified 17 genes that were downregulated with |log2FoldChange| ≥ 2. Through BLASTP searches, we identified the following homologous genes in yeast: Rrp5, an assembly factor responsible for the assembly of both the pre-60S and pre-40S ribosomal subunits [37,38]; PAK1IP1 orthologs exhibit essentiality for cell viability and involvement in 60S rRNA biogenesis [39]; U3 snoRNA, the most conserved and abundant snoRNA, playing a crucial role in processing the primary pre-rRNA transcript [40,41,42]. These findings are consistent with our previous speculation that the use of validamycin affects the ribosome synthesis process by significantly downregulating ribosome assembly factor-related genes, which, in turn, affects the growth of fungi. Kasugamycin and Cycloheximide, produced by *S. kasugiensis* and *S. griseus*, respectively, are commonly used for plant protection. Kasugamycin inhibits protein synthesis in bacteria by binding to the 30S subunit of the bacterial ribosome, whereas Cycloheximide acts by binding to the 60S subunit of the eukaryotic ribosome and inhibits protein synthesis in fungi [43,44,45]. These results suggest that ribosome assembly factors may have some sensitivity to aminoglycoside antibiotics, thereby affecting the protein synthesis process during fungal infection. Validamycin, produced by *Streptomyces*, is also an aminoglycoside antibiotic, but its impact on protein synthesis in fungi is not fully understood.

In this study, significant impacts on the MAPK signaling pathway (Figure 4b), protein translation, transport, and metabolism, as well as signal transduction pathways, were revealed by transcriptome data analysis (Figure 4c). Meanwhile, we found that most of the genes involved in ribosome biogenesis were significantly inhibited (Figure 4c and Figure 5). Combined with GO analysis, genes related to rRNA processing and maturation, ribosome self-assembly, and protein synthesis were all affected (Figure 3d). It is speculated that validamycin may also lead to large-scale inhibition of downstream gene expression by inhibiting key upstream key signaling pathways regulating ribosome formation and assembly. This study provides a new clue to further clarify the mechanism of validamycin in fungi.

## Figures and Tables

**Figure 1 jof-09-00846-f001:**
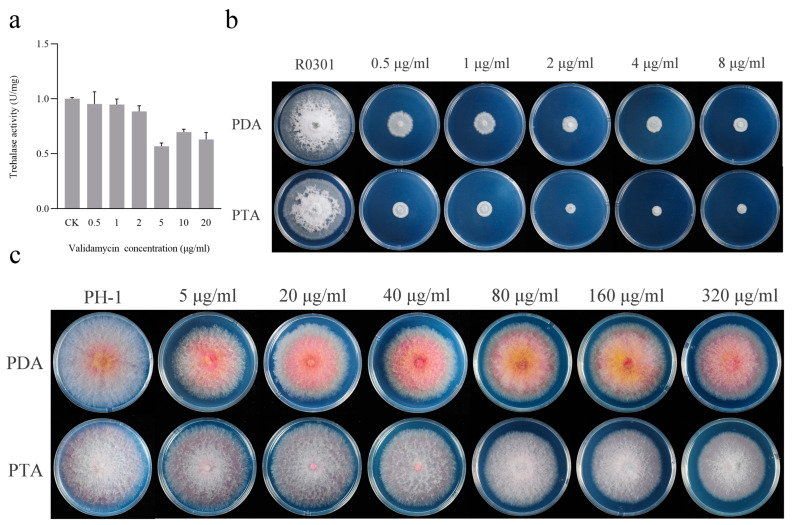
Sensitivity of the strain R0301 to validamycin. (**a**) Validamycin inhibited trehalase activity in the mycelium of *R. cerealis*. (**b**) Validamycin inhibited the hyphal growth of *R. cerealis* on PDA and PTA plates (25 °C, 3–8 days). The concentration of validamycin was set to 0, 0.5, 1, 2, 4, and 8 μg/mL. (**c**) *F. graminearum* was treated at 0, 5, 20, 40, 80, 160, and 320 μg/mL under the same conditions.

**Figure 2 jof-09-00846-f002:**
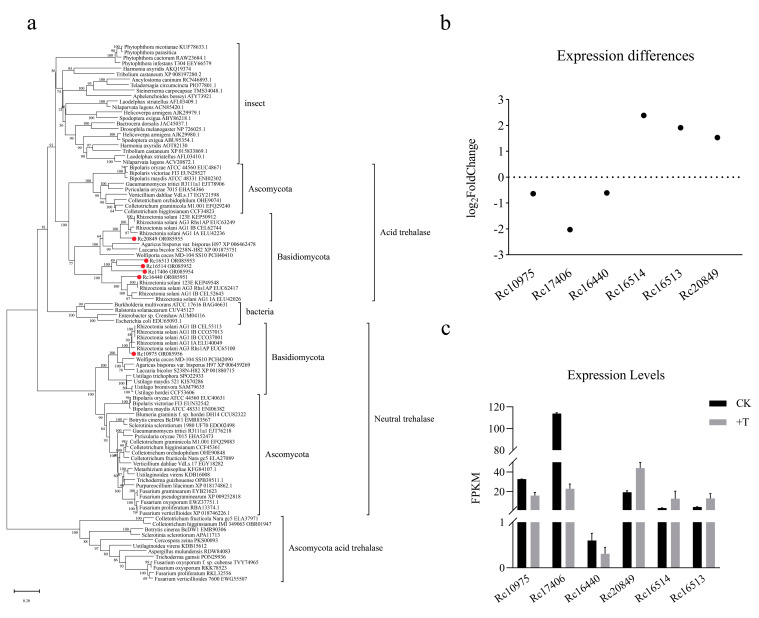
Phylogenetic tree analysis of trehalase proteins. (**a**) The phylogenetic tree based on the amino acid sequence was constructed by the MEGAX program using the maximum likelihood method. The amino acid sequences from different species were aligned by MAFFT. (**b**) Expression differences and (**c**) expression levels of transcriptome data from *R. cerealis*.

**Figure 3 jof-09-00846-f003:**
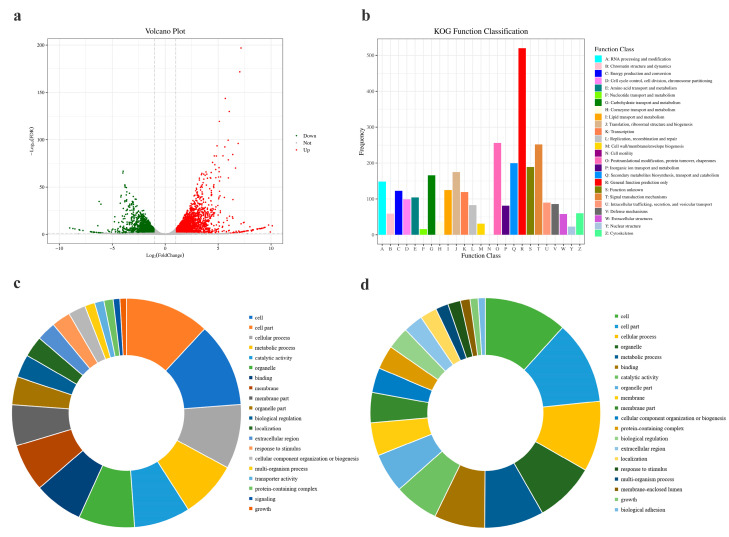
Transcriptome data analysis of trehalase. (**a**) Volcano plot showing the distribution of DEGs at |log2(fold-change)| ≥ 2 and FDR (false discovery rate) *p* < 0.05 before and after validamycin treatment. The red dots in the plot represent the statistically significant upregulated DEGs, the green dots represent the statistically significant downregulated DEGs, and gray dots indicate the DEGs without statistical significance. (**b**) KOG classification of DEGs. (**c**) Pie charts showing the top 20 upregulated and (**d**) downregulated DEG enriched GO terms.

**Figure 4 jof-09-00846-f004:**
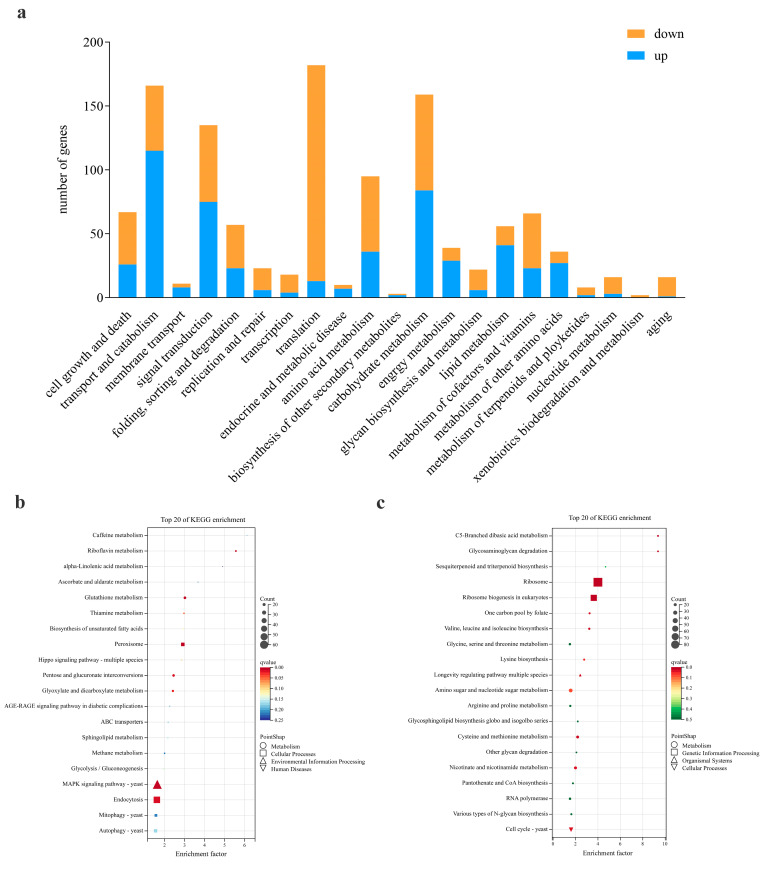
Kyoto Encyclopedia of Genes and Genomes (KEGG) functional annotations of DEGs in R0301 vs. R0301 treated with validamycin. (**a**) Accumulated bar chart indicates the number of genes enriched by each KEGG pathway. Blue represents genes that are upregulated and orange represents genes that are downregulated. (**b**) Bubble charts showing the top 20 DEGs that are upregulated or (**c**) downregulated before and after validamycin treatment that are enriched in the KEGG pathway. The color of the bubble represents the corrected *q* value. The size of the bubble indicates the number of DEGs. The KEGG pathways are listed according to their molecular interactions and reaction networks within the ‘metabolism’, ‘genetic information processing’, ‘environmental information processing’, ‘cellular processes’, ‘human diseases’ and ‘organismal systems’ categories with different types of shapes.

**Figure 5 jof-09-00846-f005:**
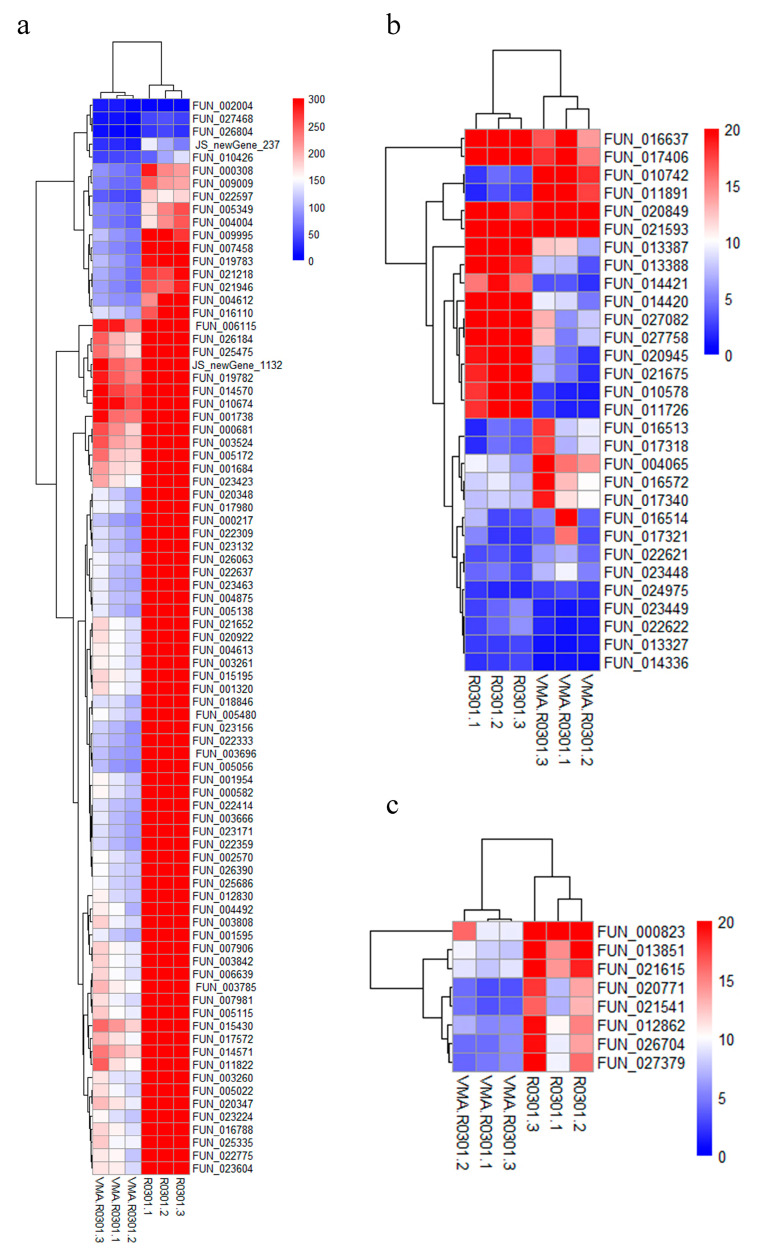
Expression values of related pathway genes before and after treatment with validamycin. (**a**) Heatmap of the expression values of genes related to starch and sucrose metabolism. (**b**) Heatmap of expression values of genes related to ribosome biogenesis in eukaryotes. (**c**) Heatmap of the expression values of genes related to ribosomes. Red indicates that the expression was relatively high, while blue indicates that the expression was relatively low. White indicates that the gene expression between the conditions analyzed was unchanged.

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
