# Peer review of "Trehalase Inhibitor Validamycin May Have Additional Mechanisms of Toxicology against *Rhizoctonia cerealis"

_jof, 2023, doi:10.3390/jof9080846_

Round 1
Reviewer 1 Report
The manuscript presents some new evidences that trehalase inhibitor validamycin may have additional mechanisms of toxicology against Rhizoctonia cerealis. However, there are also some problems and errors, including English grammar and academic issues, in the manuscript that need to be revised.
Main comments:
P1L6: Jiangsu, Nanjing 210014” should be changed to “Nanjing 210014, Jiangsu”.
P1L8: “Jiangsu, Yangzhou 225009” should be changed to “Yangzhou 225009, Jiangsu”.
P1L10: “Hubei, Jingzhou 434025” should be changed to “Jingzhou 434025, Hubei”
P1L12: Abstract. The effective results and information of the abstract are insufficient. It is suggested to supplement some detailed transcriptome data.
P1L12-13: In order to be parallel to wheat, barley should also be noted with its Latin scientific name.
P1L13: “fungi” should be changed to “fungus”.
P1L21: “indicate” should be changed to “indicated”.
P1L26: Suggest changing “the treatment group given validamycin” to “the validamycin treatment group”.
P1L33: “Sharp eyespot” should be changed to “Sharp eyespot of wheat”.
P1L38: “anti-sharp eyespot wheat breeding” should be changed to “breeding for resistance varieties against sharp eyespot of wheat”.
P2L55: Fusarium should be italic.
P2L55: FHB should be full spelling for its first time appearance.
P2L78: “potato liquid medium” might be “potato dextrose broth (PDB) medium”?
P3L99: “the OD40 nm” should be changed to “the optical density at 540 nm (OD540)” .
P3L108: “established” should be changed to “constructed”.
P3L114: “mycelium was” should be changed to “mycelia were”.
P3L134: in vitro should be italic.
P3L136: What is the meaning of CK? It should be explained in "Materials and Methods".
P3L140: “morphology” should be changed to “morphologies”.
P3L141: “was” should be changed to “were”.
P4L148: “R0301 strain” should be changed to “strain R0301”.
P5L188: “DEG” should be full spelling for its first time appearance.
P8L245: “DON” should be full spelling for its first time appearance.
P8L254-256: “Although there are reports that validamycin reduces the progression of the disease by inhibiting trehalase activity, its target or mechanism of action is still unclear”, the font size of this paragraph is significantly larger than other texts.
P9L260-262: “and the expression level of Rc17406 was most downregulated, suggesting that this gene may play a major role in the metabolism of trehalose in R. cerealis (Fig. 2b, 2c)”, the font size of this paragraph is significantly smaller than other texts.
P9L264, 267: “validamycin” smaller than others.
P9L275-311: The font size of this paragraph is significantly smaller than other others.
The manuscript contains some grammatical errors.
Reviewer 2 Report
Validamycin is a trehalase inhibitor as reported in many publications. However, other effects of validamycin are still unknown. This nice, extensive report revealed to us that validamycin had many aspects, more than just an enzyme inhibitor. Validamycin has different effects in different fungi. This report is also one of the best example of the use of validamycin in the future. The methods and results of this report are well prepared and clear to me. The discussion part are very extensive in mechanisms and application including future directions.
Author Response
We appreciate the positive feedback provided by the reviewer 2.
Reviewer 3 Report
My main concern is that the presence and concentration of trehalase after precipitation was not measured and documented. by A280 or electrophoresis ... Enzyme acitivity depends on the concentration...also additional components if trehalase is not purified could influence results. Also, for me is not clear final aim of research and final outcome info for the reader. I am not convinced that authors clarified and get known mechanism of validamycin action. Importantly quality of images is very low. Additional comment are included in the attached manuscript pdf. Without adequate correction manuscript should not be accepted.

Manuscript should be rewritten to be more fluent and readable.
Some comments are written in attached pdf
Round 2
Reviewer 3 Report
Manuscript was corrected by authors leading to its better quality. Image with phylogenetic tree is still difficult to read, but I understand difficulties with presenting so many data. Maybe presentation of this image as a separate supplementary material file would help to resolve this?
Author Response
We appreciate the reviewer's thoughtful comments. We will upload Fig2 and phylogenetic tree images separately to ensure that the readers can see more accurate information.
